# Assessing trends and reasons for unsuccessful implant discontinuation in Burkina Faso and Kenya between 2016 and 2020: a cross-sectional study

Katherine Tumlinson ![ORCID],[1,2] Leigh Senderowicz,[3,4] Brooke W Bullington ![ORCID],[2,5] Stephanie Chung,[1] Emilia Goland,[1] Linnea Zimmerman ![ORCID],[6] Peter Gichangi,[7] Mary Thiongo,[7] Georges Guiella,[8] PMA Principal Investigators Group, Celia Karp[6]

For numbered affiliations see end of article.

**Correspondence to**
Brooke W Bullington;
bbullington@unc.edu

## ABSTRACT

**Objectives** Contraceptive implant use has grown considerably in the last decade, particularly among women in Burkina Faso and Kenya, where implant use is among the highest globally. We aim to quantify the proportion of current implant users who have unsuccessfully attempted implant removal in Burkina Faso and Kenya and document reasons for and location of unsuccessful removal.

**Methods** We use nationally representative data collected between 2016 and 2020 from a cross-section of women of reproductive age in Burkina Faso and Kenya to estimate the prevalence of implant use, proportion of current implant users who unsuccessfully attempted removal and proportion of all removal attempts that have been unsuccessful. We describe reasons for and barriers to removal, including the type of facility where successful and unsuccessful attempts occurred.

**Findings** The total number of participants ranged from 3221 (2017) to 6590 (2020) in Burkina Faso and from 5864 (2017) to 9469 (2019) in Kenya. Over a 4 year period, the percentage of current implant users reporting an unsuccessful implant discontinuation declined from 9% (95% CI: 7% to 12%) to 2% (95% CI: 1% to 3%) in Kenya and from 7% (95% CI: 4% to 14%) to 3% (95% CI: 2% to 6%) in Burkina Faso. Common barriers to removal included being counselled against removal by the provider or told to return a different day.

**Conclusion** Unsuccessful implant discontinuation has decreased in recent years. Despite progress, substantial numbers of women desire having their contraceptive implant removed but are unable to do so. Greater attention to health systems barriers preventing implant removal is imperative to protect reproductive autonomy and ensure women can achieve their reproductive goals.

## STRENGTHS AND LIMITATIONS OF THIS STUDY

⇒ This analysis uses nationally representative Performance Monitoring for Action data from multiple years across diverse geographies.
⇒ Survey items were asked inconsistently between 2016 and 2020, making direct comparisons difficult in some instances. For example, comparison of reasons and barriers for implant removal was difficult to compare over time.
⇒ Small sample sizes of women seeking implant removal limited greater assessment of unsuccessful removal experiences.
⇒ Data do not capture implant removal experiences of women who switched contraceptive methods.

commodity cost of implants. This has resulted in expanded access to implants in LMICs, diversifying the contraceptive method mix and contributing to substantial growth in the contraceptive prevalence rate, particularly in sub-Saharan Africa.[1 2] Implants, a form of long-acting reversible contraception (LARC), are highly effective, easy to maintain and relatively low cost over time, making them a popular choice for many women in sub-Saharan Africa.[3 4] In Burkina Faso and Kenya, patterns in contraceptive use have changed dramatically in recent years, as many women have shifted from using injectable contraception to implants.[2 5] In both countries, implants are heavily subsidised by the government and donors, and are provided free of charge or at a low cost.[6 7]

While contraceptive implants offer many advantages, insertion and removal require a trained clinician, posing potential barriers to successful uptake and discontinuation. To uphold principles of human rights, provision of reliable, sustainable and equitable services for both insertion and removal of implants must be ensured. Lack of access

## BACKGROUND

Access to contraceptive implants has increased substantially in many low-income and middle-income countries (LMICs) over the past decade. Political and financial commitments to contraceptive access from governments and international donors has led to increased supply and reduced

to high-quality implant removal services compromises reproductive autonomy, particularly for individuals who wish to stop using the method.[8] With increased uptake of implants, the demand for removal service in sub-Saharan Africa is estimated to increase substantially in the coming years, yet the extent to which individuals can access such services, as well as barriers they may face in seeking removal, has not been readily researched.[9] Additionally, understanding the unique sociodemographic characteristics of implant users can provide important insights into the potential implications of such barriers.

A 2019 qualitative study shed light on several barriers to person-centred care, specifically for individuals receiving LARC methods. Women from a sub-Saharan African country described being pressured to adopt LARC they did not want or fully understand due to limited choices, biased counselling, scare tactics and misinformation from service providers. Several women reported being denied removal services before completing the 5 year contraceptive efficacy period for the method, despite their desires for removal, even if they wished to become pregnant.[10] Similar findings emerged in a qualitative study from Western Kenya, where women shared experiences of being refused removal and pressured to use LARC for its full duration, regardless of their reported side effects. Other barriers to removal services included high removal costs, both in terms of formal and informal fees.[11] In a mixed-method study among women in Burkina Faso and Uganda, respondents reported anxiety over the potential added costs of method removal.[12] While qualitative evidence demonstrates barriers women face when seeking removal, few quantitative studies have estimated this experience among population-based samples of current and recent implant users, limiting our understanding of the frequency and reasons for unsuccessful removals.

This study seeks to expand on recent qualitative findings highlighting substantial barriers to implant removal. Using nationally representative data collected between 2016 and 2020 in Burkina Faso and Kenya, we examine sociodemographic characteristics of implant users, recent trends in implant use, unsuccessful removal and barriers to successful removal. We aim to quantify the proportion of current implant users who have unsuccessfully attempted implant removal in each country, and document reasons for and location of unsuccessful removal over the 4 year period.

## METHODS
### Data source and sample
This cross-sectional study uses multiple rounds of survey data from the Performance Monitoring for Action (PMA) project, formerly known as PMA2020. Since 2019, PMA has conducted yearly population-based surveys with women of reproductive age (15–49 years), in 10 countries across sub-Saharan Africa and South Asia, generating data on key reproductive health indicators. Data from PMA2020 on implant removal are available in selected

countries dating back to 2016. We restrict our analysis to the two countries that measured implant removal experiences across multiple rounds of data from PMA2020 and PMA, and for which implants constitute the most widely used method. Based on these criteria, we include data collected in Burkina Faso and Kenya between 2016 and 2020, with each country contributing four rounds of data. Participating women were selected using a multistage sampling strategy. All eligible women were invited to participate via an informed consent process. Enumerators collected data from each participant, including background characteristics, reproductive and contraceptive behaviours and experiences with contraceptive health services (eg, implant removal). Additional details of PMA's survey methodology have been published elsewhere and can be found at https://www.pmadata.org/data/survey-methodology.[13]

### Patient and public involvement
Patients and the public were not involved in the design, conduct, reporting, or dissemination plans of our research.

### Measures
Our primary measures assess women's contraceptive use and, among users of the contraceptive implant, experiences with discontinuation. To assess changes in contraceptive use, implant use and the contraceptive method mix since 2016, we used women's responses to the question, 'Are you or your partner currently doing something or using any method to delay or avoid getting pregnant?' Among women who reported use, they were asked to specify which method they were currently using. We compare use of the implant to use of other modern methods, defined as female or male sterilisation, intrauterine device, injectable contraception, contraceptive pill, external or internal condom, emergency contraception, standard days method, or the lactational amenorrhea method.[14 15]

Our main outcome was unsuccessful implant discontinuation, measured by asking current implant users if they have tried to have their current implant removed in the last 12 months (yes/no). Current implant users who reported having tried to have their implant removed were categorised as experiencing an 'unsuccessful' removal. Our secondary outcomes were reasons for and location of unsuccessful removal, which were ascertained by asking those with unsuccessful removal why they were not able to have their implant removed and where they went or who attempted to remove their implant. Responses to the latter were categorised as a public (ie, government owned) or private-sector facility/provider.

Additionally, we estimated successful implant removal among women who were not currently using contraception and had used a method in the last 12 months. Current non-users reporting implant use within the last 12 months were classified, de facto, as having successful implant removal. Due to limitations of the existing data

(ie, absence of a contraceptive calendar in selected years in both countries), we are unable to estimate the proportion of successful implant removal among *currently* contracepting women (ie, those women who removed an implant in the last 12 months and were using a different method at the time of data collection); we describe this limitation in further detail in the discussion section. Among women with a successful implant removal in the last 12 months, we also ascertained reasons for and location of successful implant removal, also delineating reliance on the public or private sector for this service. In the first three rounds of data included in this analysis (annually, 2016–2019), reason for removal was measured directly by asking women who reported implant removal in the last 12 months 'Why did you stop using your implant?' For the most recent round of data collection in each country, this question was omitted, and both implant removal and reasons for removal were assessed using data from the contraceptive calendar (restricted to the 12 months preceding the interview). The contraceptive calendar is a retrospective instrument that documents duration of use and reasons for discontinuation, on a month-by-month basis, typically for multiple years.

Notably, not all measures related to implant removal were collected for all years in both countries. The location of successful removal was only captured for 2 years for each country (Burkina Faso: 2018 and 2019; Kenya: 2017 and 2018) and the location of unsuccessful removal was not collected in Burkina Faso in 2017 or Kenya in 2016. Additionally, reason for unsuccessful implant discontinuation was not collected in Kenya in 2017. Despite these limitations, we use all available data in each geography to examine trends and barriers to removal.

### Analysis

We first describe sociodemographic characteristics of modern contraceptive users in each country for the most recent year of data. We use bivariate statistics and Pearson's $\chi^2$ tests to explore whether the use of the implant versus any other modern method of contraception varied according to women's sociodemographic characteristics. To contextualise trends in access to removal, for each country and survey year, we describe the proportion of women currently using a contraceptive method, those using a modern method and the overall contraceptive method mix.

Next, we estimate the proportion of current implant users who reported unsuccessful removal. We also estimate the proportion of all implant discontinuation attempts that were unsuccessful by dividing the total number of unsuccessful discontinuation attempts in the past 12 months by all discontinuation attempts in the past 12 months, both successful and unsuccessful. We use descriptive statistics to summarise locations of and reasons for successful and unsuccessful removal. Analyses were computed using Stata V.16.0. Survey weights were applied to account for the probability of selection as well as non-response.

Ethical approval for PMA data collection activities was granted by in-country ethics committees in Burkina Faso (Comité d'Ethique pour la Recherche en Santé) and Kenya (Kenya Medical Research Institute (KEMRI) Ethics Review Committee). The Institutional Review Board for human subjects research at the Johns Hopkins Bloomberg School of Public Health reviewed and approved the study protocols for the PMA project.

## RESULTS

Online supplemental table 1 provides background characteristics for all modern contraceptive users in the most recent year of data collection, stratified by use of implants vs other modern methods. Implants users, compared with other modern method users, were more likely to be older, married, less educated, live in rural areas and poorer households, have a greater number of children and obtain their method from the public sector. While implant users in Kenya were more likely to prefer no more children compared with other modern method users (48% vs 44%, respectively), this difference was small, and no statistically significant differences were seen in the distribution of fertility preferences in Burkina Faso.

In online supplemental table 2, we provide important context for trends in implant removal by presenting the prevalence of all contraception, modern contraception and the method mix in each country for each year of data collection. The total number of participants ranged from 3221 (2017) to 6590 (2020) in Burkina Faso and from 5658 (2018) to 9469 (2019) in Kenya. In Burkina Faso, contraceptive prevalence increased in the first 3 years, from 23% in 2017 to 29% in 2019, while contraceptive prevalence remained stable at 45% in Kenya during this time.

Figure 1A,B display trends in implant and injectable use. In Burkina Faso, implant use constituted 43% of the method mix in both 2017 (95% CI: 38% to 49%) and 2020 (95% CI: 39% to 48%) but the gap between implants and injectables (the second most common method) grew, as injectable use declined from 30% (95% CI: 23% to 39%) to 24% (95% CI: 19% to 31%). In Kenya, implant use grew from 28% (95% CI: 25% to 30%) in 2016 to 36% (95% CI: 34% to 38%) of the method mix in 2019. Due to a corresponding large decline in injectables (42% to 33%), implants are roughly tied with injectables as the primary contraceptive method in Kenya.

Figure 2A illustrates the percent of current implant users reporting an unsuccessful discontinuation attempt, and figure 2B presents the percent of all implant discontinuation attempts that have been unsuccessful. In Burkina Faso, the percent of current implant users who reported an unsuccessful discontinuation attempt in the last 12 months declined from 7% (95% CI: 4% to 14%) in 2017 to 3% (95% CI: 2% to 6%) in 2020. In Kenya, the percent of current users reporting unsuccessful removal dropped from 9% (95% CI: 7% to 12%) in 2017 to 2% (95% CI: 1% to 3%) in 2019. Among all discontinuation

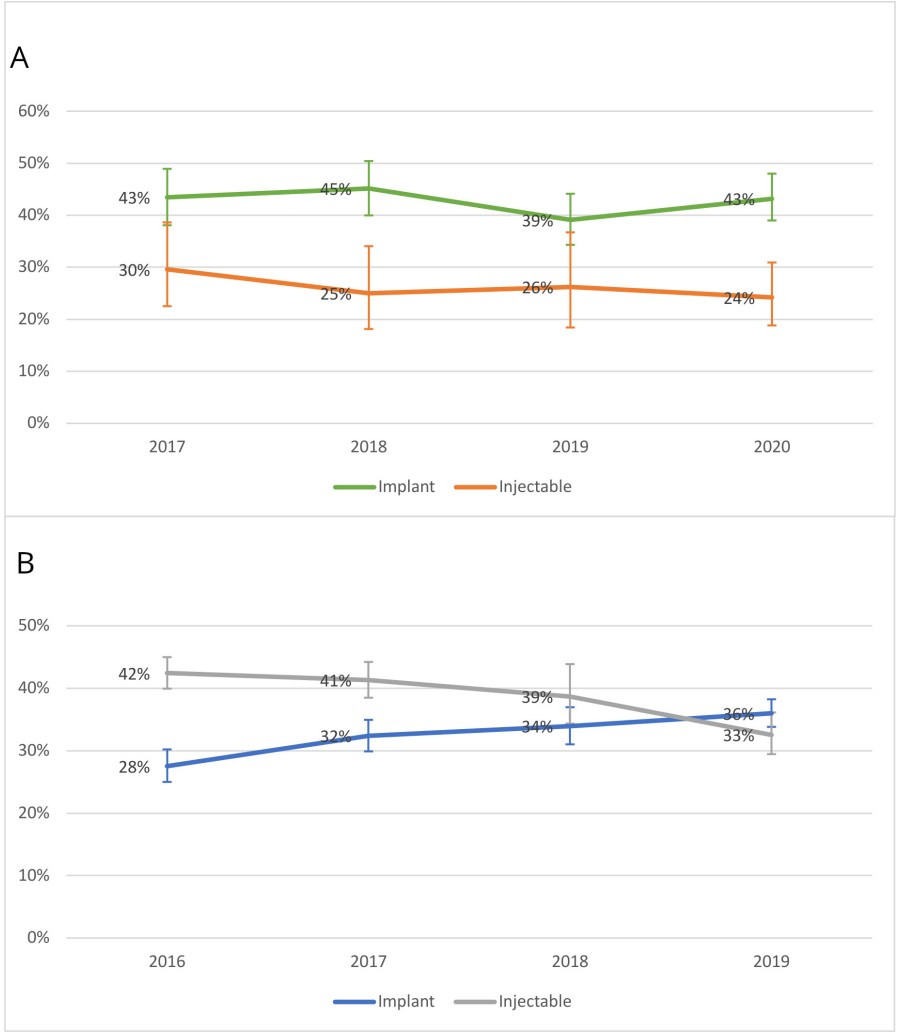

**Figure 1** Implants and injectables as a percent of the total modern contraceptive method mix: (A) Burkina Faso and (B) Kenya.

attempts in Burkina Faso, the percent of removals that were unsuccessful dropped from 28% (95% CI: 19% to 40%) in 2017 to 15% (95% CI: 10% to 22%) in 2020. In contrast, Kenya dropped from 52% (95% CI: 44% to 61%) in 2017 to 13% (95% CI: 9% to 18%) in 2019.

Among women who had successfully removed their implant, we present reasons for discontinuation in Burkina Faso (figure 3A) and Kenya (figure 3B). In Burkina Faso, reasons for discontinuation varied over time, with women in 2016 most frequently reporting removal due to a desire to become pregnant (60%) or concerns about interference with the body's natural process (24%). Although these reasons were still commonly reported in later survey years, the proportion of women reporting these experiences diminished over time. By 2020, concerns over side effects were the primary reason for removal—reported by 42% of women—although more than one in three women still sought removal due to a desire to become pregnant. Method failure (became pregnant) was a frequent reason for removal in 2018 and 2019 but was reported by few women in 2017 and 2020. Very similar patterns were observed in Kenya, with a desire to become pregnant and health concerns/side effects consistently reported by

a large percent of respondents across all four rounds of data collection (29%–49% and 27%–48%, respectively), and up to half (42%–51%) reporting method failure in 2017–2018.

Figure 4 presents the location of implant removal and discontinuation attempts, stratified by public versus private sector. Among Burkinabé women, removals were most frequently sourced from a public-sector facility for all years, for both successful removals (75%–83%) and unsuccessful removals (85%–90%). In Kenya, women sourced successful removal equally from both public and private sector locations in 2017 but far more commonly from the public-sector (79%) in 2018. Like Burkina Faso, most unsuccessful removals in Kenya were sourced from the public sector in all years (71%–90%).

Among Burkinabé women who were unsuccessful in obtaining implant removal, self-reported barriers to removal fluctuated from year to year (figure 5A). In 2017, Burkinabe women primarily reported that they were counselled against removal (23%), or a trained provider was unavailable (19%). The following year, women reported a provider refused to remove their implant (18%) or that they could not afford the cost of removal (28%). Cost

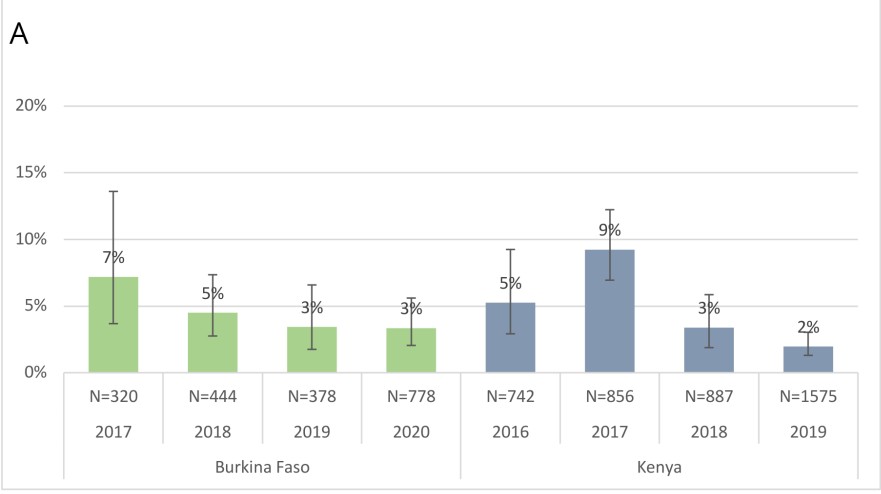

A

*Denominator is all current implant users.

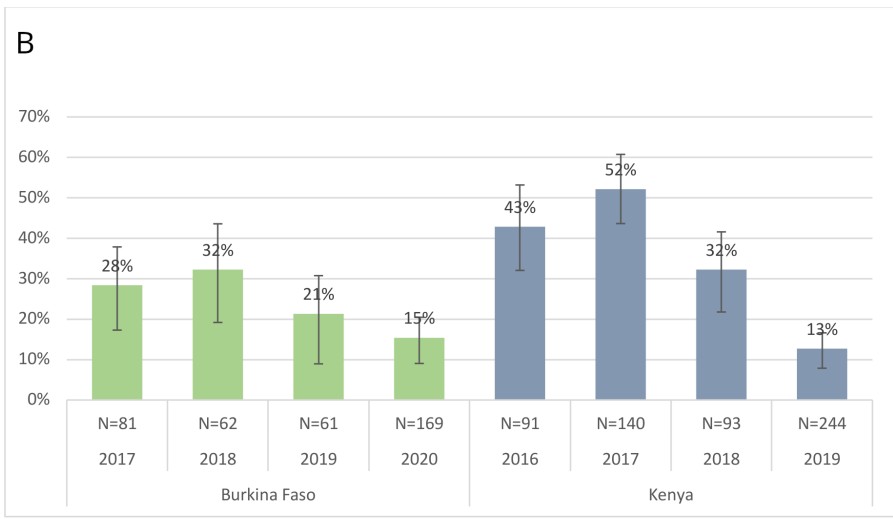

B

*Denominator is women with successful implant removal in the last 12 months and women with unsuccessful implant removal in the last 12 months. Due to limitations of the existing data, we are unable to estimate the proportion of successful removal among currently contracepting women.

**Figure 2** (A) Percent of current implant users reporting unsuccessful discontinuation attempt. (B) Percent of all discontinuation attempts that were unsuccessful.

remained a dominant barrier in 2019 but by 2020, nearly half of women (43%) reported they were counselled against removal, while more than one-third (36%) shared they were told to return to the facility another day. Similarly, in Kenya, common self-reported barriers shifted from year to year, with the highest proportion of women in 2016 and 2019 reporting a lack of a trained provider or cost, while in 2018 nearly half of women were counselled against removal or instructed to return another day (figure 5B).

## DISCUSSION

In light of growing implant use and recent qualitative research documenting barriers to implant removal in sub-Saharan Africa, we explored changes over time in contraceptive use, implant use and experiences with implant removal in Burkina Faso and Kenya. Over a 4 year

period, contraceptive prevalence was relatively steady in both settings. As a percentage of the overall method mix, implant use in Burkina Faso did not vary substantially from 2017 to 2020 (remaining around 43%), while injectable use occupied an *increasingly* distant second place. In contrast, Kenya experienced a steady increase in implant use, coupled with commensurate declines in injectable use.

Given the predominance of implants in both settings, access to method removal is imperative. We found that the percent of current implant users reporting an unsuccessful discontinuation attempt declined in both countries. Reasons for this trend are unclear. Given more than one out of every 10 women of reproductive age in each country is currently using a contraceptive implant, the 2%–3% of current users unable to remove their implant translates into tens of thousands of women who

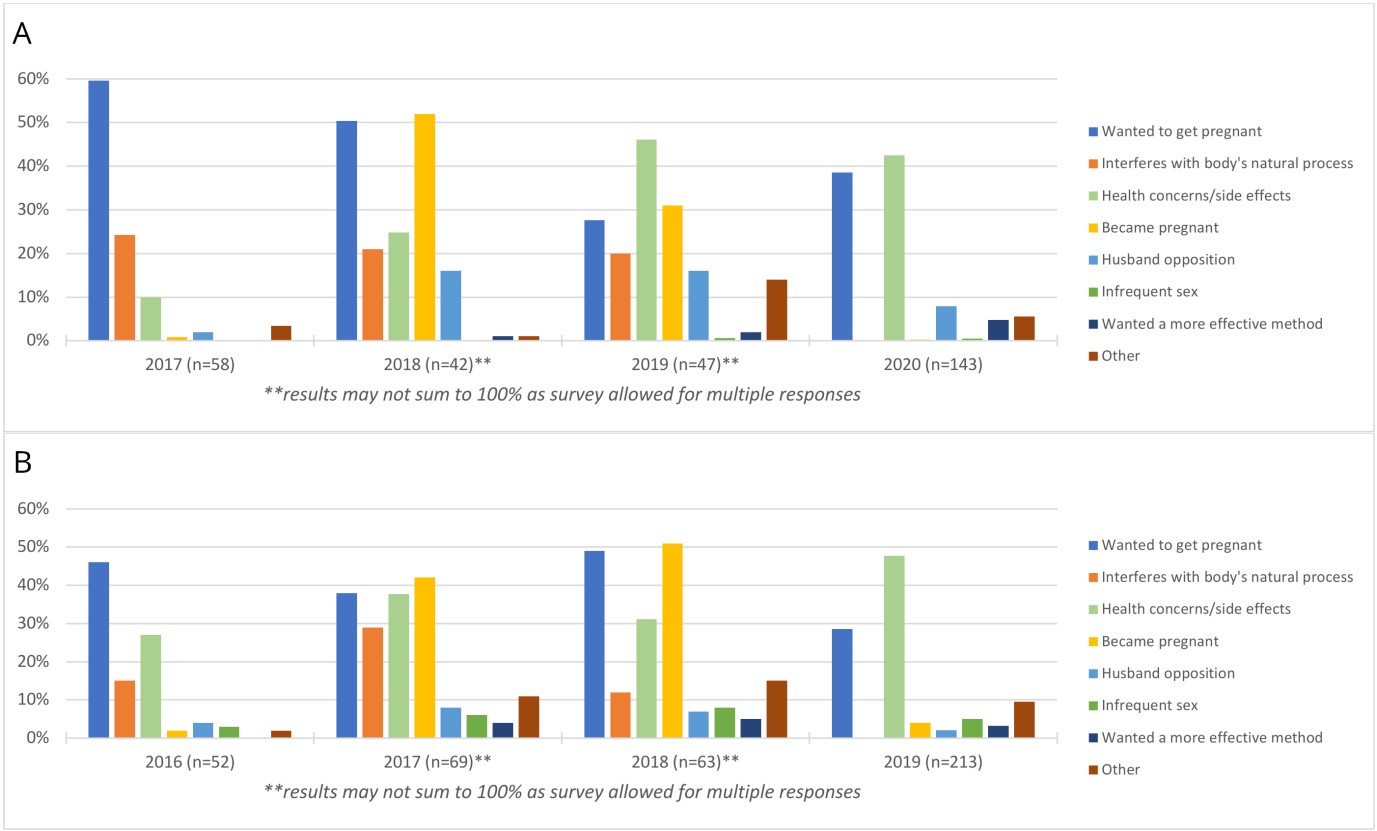

**Figure 3** Reasons for implant discontinuation: (A) Burkina Faso and (B) Kenya.

experience structural forms of contraceptive coercion that prevent them from achieving their contraceptive goals.

In both Burkina Faso and Kenya, more than one in eight women seeking implant removal were unsuccessful in their attempt. In Kenya, this is evidence of a stark downward trend from one in two women just 3 years prior—suggesting considerable progress has been made to help

reduce barriers to removal in Kenya; yet more progress is needed. The most common reasons women seek removal changed over time in both countries; however, wanting to get pregnant or becoming pregnant and concerns about the implant's effect on the body and side effects were persistent reasons for desired removal in both countries. When women provided reasons for unsuccessful implant discontinuation, facility-level barriers were common,

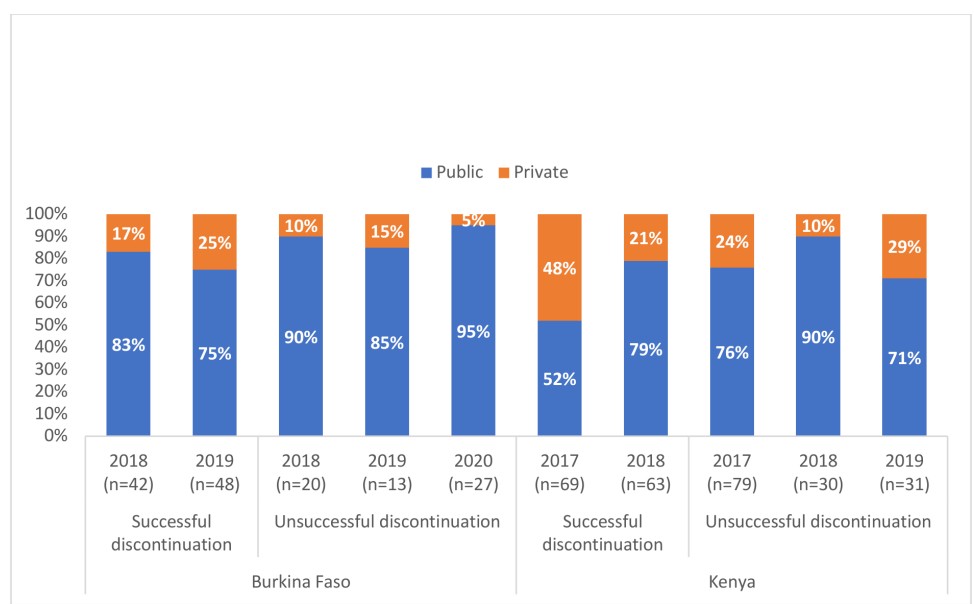

**Figure 4** Location of implant discontinuation and discontinuation attempt in Burkina Faso and Kenya, 2017–2019.

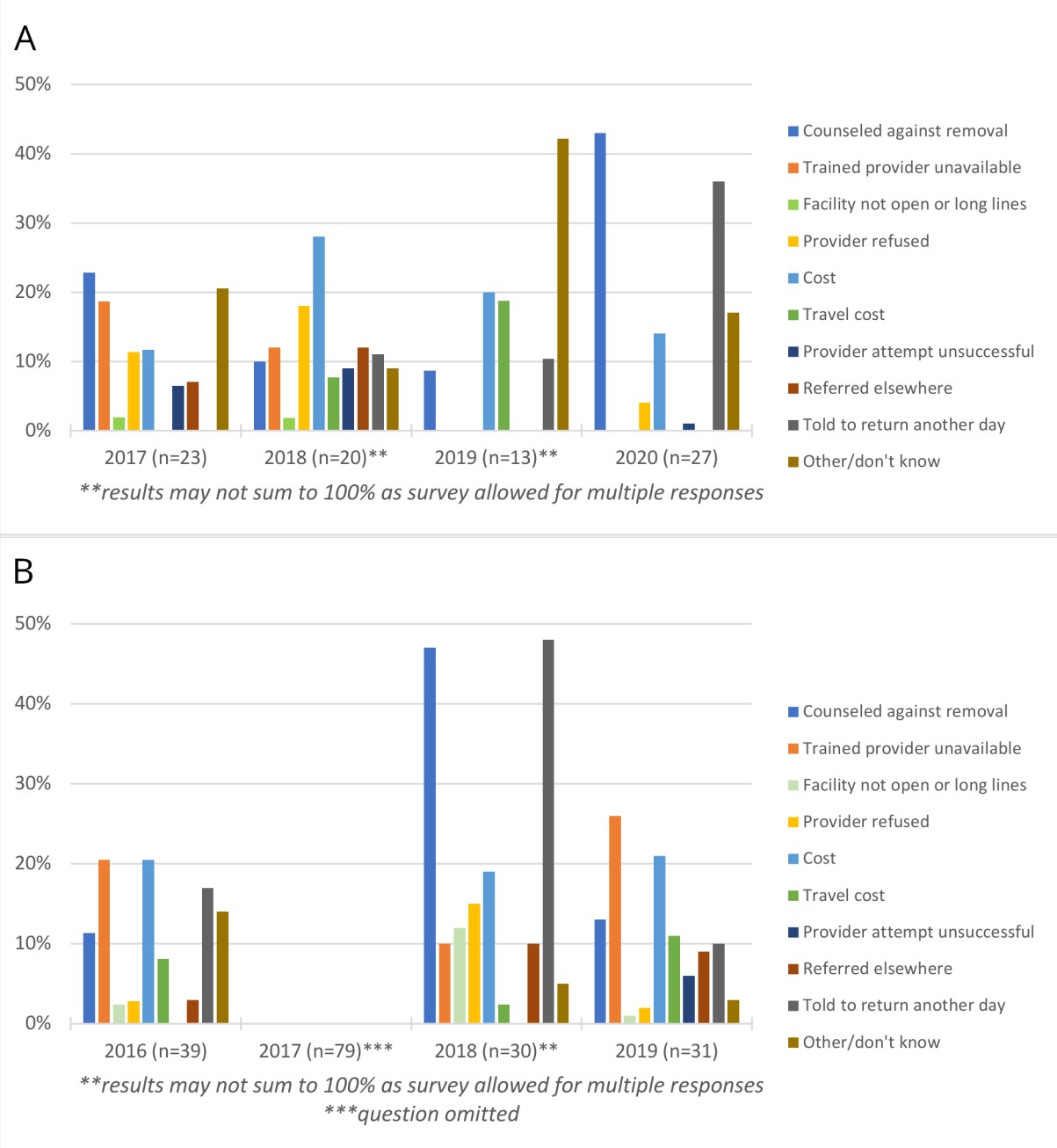

**Figure 5** Barriers to implant discontinuation: (A) Burkina Faso and (B) Kenya.

including being counselled against removal, told a trained provider is unavailable or asked to return another day. Cost was also a persistent barrier across time in both countries. Our results confirm prior findings that women encounter substantial facility-level barriers to implant removal as well as unaffordable removal fees, including within public facilities.[10 11 16–18]

Contraceptive implant users—compared with all other modern method users—were consistently older and more likely to be married, which may confer some degree of agency. However, they were also more likely to be uneducated, poor and living in rural locations, where method removal may rely heavily on public-sector facilities. Women in both countries largely source implant

removal—both successful and unsuccessful—from the public sector, where a nascent body of literature suggests that providers are resistant to implant removal—at times to preserve limited implant supplies or sometimes due to an aversion for the procedure itself. Additionally, when noting the combination of cost as a substantial barrier and public facilities as the source, we posit a system of substantial informal fees for implant removal may enable providers to avoid removing implants, particularly among poor public-sector family planning clients who are especially price-sensitive. Informal payments are unofficial financial payments from patients to providers, over and above the official cost of services. Systems of informal payments are well

documented in several countries in sub-Saharan Africa, including Kenya.[11 19]

This analysis benefits from multiple years of nationally representative data collected in diverse geographies. However, we acknowledge several limitations. First, we are limited by inconsistent survey items across all years of data. For example, in some years and for questions related to reasons for (or barriers to) implant removal, multiple responses were allowed, while in other years a single response option was used. This makes direct comparison of the frequency of reasons and barriers over time more difficult, as women may have multiple reasons for seeking or failing to have an implant removed but would have been unable to report all reasons in select survey years, resulting in under-reporting of secondary reasons. For example, we note a spike in reported method failure as a reason for discontinuation in two of the 4 years of data collection for each country and suspect the multiple response option provided in those years contributed to this inconsistency in reasons for removal, given the contraceptive implant has typically low failure rates. Similarly, the change in survey structure to include the contraceptive calendar in the most recent round of data collection resulted in minor differences in how women were asked about implant removal as well as under-reporting of implant discontinuation in the years preceding the calendar, during which only non-contraceptive users were asked about prior method use in the last 12 months. Second, small sample sizes for some of the analyses limited greater assessment of unsuccessful removal experiences. Third, among those with successful implant removal in the year prior to the survey, we were unable to measure whether these women first experienced a barrier to removal prior to successful removal. And, among those who were unsuccessful, we cannot report on the total number of unsuccessful attempts. This may have resulted in an underestimate of the percentage of current users that have experienced barriers to removal, presenting a conservative picture of the difficulty of implant removal. Furthermore, these data do not capture implant removal experience of contraceptive switchers. Ideally, those women who switched from the implant to another method should be classified as having a successful removal, but current users were not asked about prior use in survey rounds without a calendar. As such, for selected years we may have underestimated the number of successful removals, which would lead to an overestimation of the proportion of all removals that were unsuccessful. This is an important limitation of the currently available data on implant removal and calls for improvements to existing survey instruments for measuring access to implant removal. We are additionally limited in this analysis by a lack of contextual information. For example, it is possible that an implant patient complaining of side effects may have received counselling to help mitigate side effects and may have continued implant use with satisfaction. We would not be able to capture this nuance with population-based data.

Despite these limitations, our findings are in line with those in other studies that have examined implant removal in sub-Saharan Africa. In a study following LARC users over 1 year, 5% of implant users in Nigeria and 7% of implant users in Zambia reported wanting their implants removed, but still using their method.[20] Another study in Ghana noted that only 55%–61% of implant users who attempted removal successfully obtained removal on their first attempt.[21] These studies also found similar reasons for wanting implants removed, including side effects and desire for more children, and similar reasons for implant removal failure, including provider counselling to continue using the method.[20 21]

This analysis provides an important contribution to the literature on barriers to implant removal in sub-Saharan Africa. Using nationally representative data to assess changes over time in barriers to removal across two distinct countries, we note both much needed reductions in these barriers and also highlight the additional progress required to sufficiently safeguard contraceptive autonomy in LMICs. Our findings contribute to a growing body of literature highlighting the imperative for improved, patient-centred implant removal services in sub-Saharan Africa.[10 11 20 21] Despite international calls to ensure programmatic support for implant removal services scaled appropriately with the popularity of the implant, barriers to implant removal services persist.[2 9] Family planning programmes and services must prioritise contraceptive autonomy, respecting a person's right to choose to discontinue their method at any time and for any reason.

### Author affiliations

[1]Department of Maternal and Child Health, Gillings School of Global Public Health, The University of North Carolina at Chapel Hill, Chapel Hill, North Carolina, USA
[2]Carolina Population Center, University of North Carolina at Chapel Hill, Chapel Hill, North Carolina, USA
[3]Departments of Gender and Women's Studies, University of Wisconsin-Madison, Madison, Wisconsin, USA
[4]Department of Obstetrics and Gynecology, University of Wisconsin-Madison, Madison, Wisconsin, USA
[5]Department of Epidemiology, Gillings School of Global Public Health, The University of North Carolina at Chapel Hill, Chapel Hill, North Carolina, USA
[6]Department of Population, Family & Reproductive Health, Johns Hopkins University Bloomberg School of Public Health, Baltimore, Maryland, USA
[7]International Centre for Reproductive Health-Kenya, Nairobi, Kenya
[8]ISSP/University of Ouagadougou, Ouagadougou, Burkina Faso

**Acknowledgements** We wish to gratefully acknowledge the support of Matt Gunther, research staff with IPUMS, for extracting and harmonising several key variables from the Performance Monitoring for Action (PMA) retrospective reproductive calendar data in Kenya and Burkina Faso. The PMA project relies on the work of many individuals, both in the USA and in survey countries. The project team is grateful for support from the Bill & Melinda Gates Foundation and would like to thank the country teams and resident enumerators who are ultimately responsible for the success of PMA.

**Contributors** Data analysis was conceived by KT, LS and CK, all of whom contributed to the analysis plan and paper framing. Quantitative analysis was conducted by BWB and KT, with some input from LS and CK. KT, BWB, SC and EG all drafted portions of the initial manuscript. CK, LZ, PG, MT, GG and the Performance Monitoring for Action (PMA) PI Group contributed to data collection, data curation,

data cleaning and project administration. KT is responsible for the overall content of the article as the guarantor. All authors contributed substantially to review and revision of the manuscript and have read and approved the final manuscript.

**Funding** This work was supported, in part, by the Bill & Melinda Gates Foundation IVN 009639. Under the grant conditions of the Foundation, a Creative Commons Attribution 4.0 Generic License has already been assigned to the Author Accepted Manuscript version that might arise from this submission. Support for this research was also provided, in part, by a career development grant (R00 HD086270) to KMT and an infrastructure grant for population research (P2C HD050924) to the Carolina Population Center at the University of North Carolina at Chapel Hill, both from The Eunice Kennedy Shriver National Institute of Child Health and Human Development (NICHD) of the National Institutes of Health (NIH). LS's contribution was supported by a Ruth L Kirschstein National Research Service Award (T32 HD049302) and Population Research Infrastructure grant (P2C HD047873) from the NICHD. BB's contribution was supported in part by funding from NICHD (T32 HD052468) and Population Research Infrastructure grant (P2C HD050924). Contributions by CK, LZ, PG, MT, GG and the PMA PI Group were supported by grants OPP1198333 and OPP1198339 awarded by the Bill & Melinda Gates Foundation.

**Disclaimer** The contents of this article are solely the responsibility of the authors and do not necessarily represent the official views of the NIH/NICHD or the Bill & Melinda Gates Foundation.

**Competing interests** None declared.

**Patient and public involvement** Patients and/or the public were not involved in the design, or conduct, or reporting or dissemination plans of this research.

**Patient consent for publication** Consent obtained directly from patient(s).

**Ethics approval** This study involves human participants and Ethical approval for Performance Monitoring for Action (PMA) data collection activities has been granted by in-country ethics committees in Burkina Faso (Comité d'Ethique pour la Recherche en Santé, #A14-2020) and Kenya (Kenya Medical Research Institute (KEMRI) Ethics Review Committee, #P241/04/2020). The Institutional Review Board for human subjects research at the Johns Hopkins Bloomberg School of Public Health reviewed and approved the study protocols for the PMA project. Participants gave informed consent to participate in the study before taking part.

**Provenance and peer review** Not commissioned; externally peer reviewed.

**Data availability statement** Data are available upon reasonable request. Access to the deidentified data sets used in this analysis are granted on a per request basis by Performance Monitoring and Action. To request data sets, please go to: https://www.pmadata.org/data/request-access-datasets

**ORCID iDs**
Katherine Tumlinson http://orcid.org/0000-0001-8314-8219
Brooke W Bullington http://orcid.org/0000-0002-3341-087X
Linnea Zimmerman http://orcid.org/0000-0002-0118-0889

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
