## [Reviewer comments · BMJ Open]

ARTICLE DETAILS

TITLE (PROVISIONAL)	Assessing trends and reasons for unsuccessful implant discontinuation in Burkina Faso and Kenya between 2016 and 2020: A cross-sectional study
AUTHORS	Tumlinson, Katherine; Senderowicz, Leigh; Bullington, Brooke; Chung, Stephanie; Goland, Emilia; Zimmerman, Linnea; Gichangi, Peter; Thiongo, Mary; Guiella, Georges; NA, PMA Principal Investigators Group; Karp, Celia

VERSION 1 – REVIEW

REVIEWER	Howett , Rebecca University of Washington
REVIEW RETURNED	03-Feb-2023

GENERAL COMMENTS	This is an excellent paper that presents a coherent analysis of implant removal data over four year periods in Burkina Faso and Kenya. The authors ground their manuscript in the current literature on implant removal in other low- and middle-income countries and the important issue of reproductive autonomy. They insightfully acknowledge the limitations of their data. It is well-written and I have only a few minor comments. 1. In line with the STROBE checklist, please state in your title and/or abstract that this was a cross-sectional study.2. Please review the p-values in Table 1. For those reported as being "0.000" or "<0.000", I wonder if this is a rounding error in Stata. They might be better reported as "<0.0001" (or other specified significance level) to show that the probability is very small but not zero and not less than zero (i.e. not negative).3. For the "Method Mix" section of Table 2, please specify if respondents were able to select more than one option.4. Regarding Figures 4 and 5, and PDF page 10 lines 36-40, you note that there was a spike in the number of implants removed due to method failure in 2018-2019 (Burkina Faso) and 2017-2018 (Kenya). Given that this is a highly effective method and that removal due to method failure was rarely reported in each country's other two years of data, please consider briefly exploring this finding in your discussion. I wonder if there is any correlation with changes to implant guidelines or training e.g. pregnancy testing requirements prior to insertion?5. Please review the single asterisks after the titles for Figures 4 and 5, they do not seem to have a corresponding explanation.
--

	6. For Figure 6, please merge the bottom row so that “Burkina Faso” is centred across all of the first five columns. 7. In the Background section (PDF page 4, lines 15-16) you mention that “In both countries, implants are heavily subsidized by the government and donors, and are provided free of charge or at a low cost.” In the Discussion section (PDF page 10, lines 54-57) you go on to “posit a system of substantial informal fees for implant removal” as the reason for cost being a barrier to implant removal”. Please consider providing more background to substantiate this and aid understanding for those who are less familiar with health systems in these two countries. For example, are the no/low costs mentioned in your Background specifically for the implant itself as a commodity, i.e. are with additional consultation fees for insertion or removal? How does “low” cost compare to income and other costs locally i.e. is even a formal “low” cost a barrier for implant users, many of whom in your data are from poorer households? Well done and I look forward to seeing the final paper.
--	---

REVIEWER	Cohen , Megan A. Emory University School of Medicine
REVIEW RETURNED	18-Feb-2023

GENERAL COMMENTS	The authors frame this study within the context of ensuring reproductive autonomy, which is an incredibly important topic. However, given significant data limitations, I would caution against drawing any conclusions regarding contraceptive coercion or failure to achieve reproductive autonomy from the results of this study. For example, there is no contextual information regarding if patients were satisfied with implant continuation after counseling regarding mitigation of side effects. Some people may be happy continuing implant use if side effects are managed, but this does not appear to be captured. There is also no contextual information regarding if patients were referred to another facility given deep or non-palpable implant and lack of facility expertise, which would be an appropriate reason to defer removal. Further, there is significant bias introduced as there is no information available for individuals who successfully underwent implant removal and switched to another contraceptive method for the majority of the study period. This is especially true for for the outcome of "percent of all removal attempts that were unsuccessful" (Figure 3. This also introduces bias into the outcome of "reasons for implant removal," since, again, a large population of individuals who may have undergone implant removal is not represented within the data. While the authors acknowledge this limitation, I do not believe they go far enough in discussing its importance in data validity and interpretation. Thus, I would recommend revision of the discussion to further elaborate on the significant data limitations and ensure that the conclusions do not go beyond what is supported by the data. It may also be helpful to estimate the percentage of current contraceptive users who switched from an implant in the last 12 months from the 2020 calendar data to contextualize the data in the other years. Additionally, "unsuccessful implant removal" or "removal attempt" as used throughout this article is confusing as it usually means an
--

	actual removal attempt that failed due to implant migration or deep implant. Recommend changing to something like "unsuccessful request for removal" or "unsuccessful implant discontinuation" throughout for clarity. It is crucial to distinguish this especially for Figure 6, as otherwise it appears that the actual success of the removal procedure may differ between public and private facilities. Other comments: Data in Table 1 and Table 2 seems discrepant. For example, Table 1 states 778 women used implant in Burkina Faso in 2020 and 900 used "All Modern" methods (which sounds inclusive of implants) but Table 2 shows implants account for only 43% of the method mix that year. Should the column header in Table 1 read "All Other Modern" methods?
--	--

VERSION 1 – AUTHOR RESPONSE

Reviewer: 1

Dr. Rebecca Howett , University of Washington

Comments to the Author:

This is an excellent paper that presents a coherent analysis of implant removal data over four year periods in Burkina Faso and Kenya. The authors ground their manuscript in the current literature on implant removal in other low- and middle-income countries and the important issue of reproductive autonomy. They insightfully acknowledge the limitations of their data.

It is well-written and I have only a few minor comments.

Author response: We appreciate the kind words and the thoughtful review of this work. We have incorporated all feedback and the manuscript is greatly improved as a result.

1. In line with the STROBE checklist, please state in your title and/or abstract that this was a cross-sectional study.

Author Response: Thank you for catching out oversight. We have updated our title and abstract to clarify that this analysis uses cross-sectional data.

2. Please review the p-values in Table 1. For those reported as being "0.000" or "<0.000", I wonder if this is a rounding error in Stata. They might be better reported as "<0.0001" (or other specified significance level) to show that the probability is very small but not zero and not less than zero (i.e. not negative).

Author response: Excellent point, thank you! We have corrected this so that these p-values read "<0.001".

3. For the "Method Mix" section of Table 2, please specify if respondents were able to select more than one option.

Author response: Thank you – we have now specified in the table footnote that the Method Mix in Table 2 includes only the most effective method, among those who listed more than one current method.

4. Regarding Figures 4 and 5, and PDF page 10 lines 36-40, you note that there was a spike in the number of implants removed due to method failure in 2018-2019 (Burkina Faso) and 2017-2018 (Kenya). Given that this is a highly effective method and that removal due to method failure was rarely reported in each country's other two years of data, please consider briefly exploring this finding in

your discussion. I wonder if there is any correlation with changes to implant guidelines or training e.g. pregnancy testing requirements prior to insertion?

Author response: Thank you for this excellent suggestion. We have added the following to the limitations paragraph in the discussion section:

“For example, we note a spike in reported method failure as a reason for discontinuation in two of the four years of data collection and suspect the multiple response option provided in those years contributed to this inconsistency in reasons for removal, given the contraceptive implant has typically low failure rates.”

5. Please review the single asterisks after the titles for Figures 4 and 5, they do not seem to have a corresponding explanation.

Author response: Thank you for catching this omission. We have removed the single asterisks for Figures 4 and 5.

6. For Figure 6, please merge the bottom row so that “Burkina Faso” is centred across all of the first five columns.

Author response: Thank you for catching this. We have corrected this formatting error for Figure 6.

7. In the Background section (PDF page 4, lines 15-16) you mention that “In both countries, implants are heavily subsidized by the government and donors, and are provided free of charge or at a low cost.” In the Discussion section (PDF page 10, lines 54-57) you go on to “posit a system of substantial informal fees for implant removal” as the reason for cost being a barrier to implant removal”. Please consider providing more background to substantiate this and aid understanding for those who are less familiar with health systems in these two countries. For example, are the no/low costs mentioned in your Background specifically for the implant itself as a commodity, i.e. are with additional consultation fees for insertion or removal? How does “low” cost compare to income and other costs locally i.e. is even a formal “low” cost a barrier for implant users, many of whom in your data are from poorer households?

Author response: Thank you so much for this important comment. We have included the following in the discussion section to better address the conversation around hypothesizing informal payments: “Informal payments are unofficial financial payments from patients to providers, over and above the official cost of services. A system of informal payments are well-documented in several countries in sub-Saharan Africa, including Kenya.”

Well done and I look forward to seeing the final paper.

Author response: Thank you so much!

Reviewer: 2

Dr. Megan A. Cohen , Emory University School of Medicine

Comments to the Author:

The authors frame this study within the context of ensuring reproductive autonomy, which is an incredibly important topic. However, given significant data limitations, I would caution against drawing any conclusions regarding contraceptive coercion or failure to achieve reproductive autonomy from the results of this study. For example, there is no contextual information regarding if patients were satisfied with implant continuation after counseling regarding mitigation of side effects. Some people may be happy continuing implant use if side effects are managed, but this does not appear to be captured. There is also no contextual information regarding if patients were referred to another facility given deep or non-palpable implant and lack of facility expertise, which would be an appropriate reason to defer removal.

Author response: Thank you for raising this potential data limitation. Regarding lack of staff trained to remove deeply placed or non-palpable implants, we would suggest that lack of such staff in a location where implants are being inserted constitutes a lack of patient-centered care. We explore this in a

separate manuscript, recently published: <https://pubmed.ncbi.nlm.nih.gov/36444203/>. However, we agree with the concern that side effects may have been addressed via counseling and we have added the following to the discussion section:

“We are additionally limited in this analysis by a lack of contextual information. For example, it is possible that an implant patient complaining of side effects may have received counseling to help mitigate side effects and may have continued implant use with satisfaction. We would not be able to capture this nuance with population-based data.”

Further, there is significant bias introduced as there is no information available for individuals who successfully underwent implant removal and switched to another contraceptive method for the majority of the study period. This is especially true for the outcome of "percent of all removal attempts that were unsuccessful" (Figure 3. This also introduces bias into the outcome of "reasons for implant removal," since, again, a large population of individuals who may have undergone implant removal is not represented within the data. While the authors acknowledge this limitation, I do not believe they go far enough in discussing its importance in data validity and interpretation. Thus, I would recommend revision of the discussion to further elaborate on the significant data limitations and ensure that the conclusions do not go beyond what is supported by the data. It may also be helpful to estimate the percentage of current contraceptive users who switched from an implant in the last 12 months from the 2020 calendar data to contextualize the data in the other years.

Author response: Thank you for highlighting the need to discuss this limitation in more detail. We agree this is an important limitation of currently available data and have added additional language to the limitations section to highlight this.

Additionally, "unsuccessful implant removal" or "removal attempt" as used throughout this article is confusing as it usually means an actual removal attempt that failed due to implant migration or deep implant. Recommend changing to something like "unsuccessful request for removal" or "unsuccessful implant discontinuation" throughout for clarity. It is crucial to distinguish this especially for Figure 6, as otherwise it appears that the actual success of the removal procedure may differ between public and private facilities.

Author response: Thank you for this helpful suggestion; we have updated our language throughout as requested.

Other comments:

Data in Table 1 and Table 2 seems discrepant. For example, Table 1 states 778 women used implant in Burkina Faso in 2020 and 900 used "All Modern" methods (which sounds inclusive of implants) but Table 2 shows implants account for only 43% of the method mix that year. Should the column header in Table 1 read "All Other Modern" methods?

Author response: Thank you for highlighting that this is confusing. Yes, this should read “All Other Modern” and we have made this change.

VERSION 2 – REVIEW

REVIEWER	Howett , Rebecca University of Washington
REVIEW RETURNED	14-Apr-2023
GENERAL COMMENTS	Thank you for making these revisions. I am happy to recommend your manuscript for publication.
REVIEWER	Cohen , Megan A.

	Emory University School of Medicine
REVIEW RETURNED	01-May-2023

GENERAL COMMENTS	Thank you for addressing the suggestions raised in the prior review. This is such an important topic, so it is equally important to ensure the data are presented and interpreted accurately. I do believe further discussing the limitations of the data has helped strengthen the manuscript. I also appreciate switching some of the "removal" language to "discontinuation" to make this more clear to the clinical reader. I would recommend that the manuscript be accepted pending just a couple minor suggestions: 1) I would recommend adding similar language to that in footnote 1 (e.g., "Due to limitations of the existing data, we are unable to estimate the proportion of successful implant removal among currently contracepting women") to the explanatory note at the bottom of Figure 2b so that this is also clear within the Figure itself. 2) Page 9 of PDF, lines 23-25: Recommend changing to either "A system of informal payments is..." or "Systems of informal payments are...". 3) I worry that Figures 5a and 5b may be difficult to read when published and wonder if they may be better suited to table format listing n (%) for each discontinuation barrier, similar to how supplemental table 2 is organized? The same may be true for Figures 3a and 3b, although that one is easier to interpret given that there are fewer reasons.
---

VERSION 2 – AUTHOR RESPONSE

Reviewer 2

Comment: I would recommend adding similar language to that in footnote 1 (e.g., "Due to limitations of the existing data, we are unable to estimate the proportion of successful implant removal among currently contracepting women") to the explanatory note at the bottom of Figure 2b so that this is also clear within the Figure itself.

Response: Thank you for this suggestion. We have added, "Due to limitations of the existing data, we are unable to estimate the proportion of successful implant removal among currently contracepting women" to the footnote of Figure 2b.

Comment: Page 9 of PDF, lines 23-25: Recommend changing to either "A system of informal payments is..." or "Systems of informal payments are...".

Response: We have updated this statement to say, "Systems of informal payments are..."

Comment: I worry that Figures 5a and 5b may be difficult to read when published and wonder if they may be better suited to table format listing n (%) for each discontinuation barrier, similar to how supplemental table 2 is organized? The same may be true for Figures 3a and 3b, although that one is easier to interpret given that there are fewer reasons.

Response: We appreciate this comment. We would like to defer to the editors on whether we update Figures 5a and 5b into table format.